# Cooking Methods and Their Relationship with Anthropometrics and Cardiovascular Risk Factors among Older Spanish Adults

**DOI:** 10.3390/nu14163426

**Published:** 2022-08-20

**Authors:** Montserrat Rodríguez-Ayala, Helena Sandoval-Insausti, Ana Bayán-Bravo, José R. Banegas, Carolina Donat-Vargas, Rosario Ortolá, Fernando Rodríguez-Artalejo, Pilar Guallar-Castillón

**Affiliations:** 1Department of Preventive Medicine and Public Health, School of Medicine, Universidad Autónoma de Madrid and CIBERESP (CIBER of Epidemiology and Public Health), 28029 Madrid, Spain; 2Department of Microbiology and Parasitology, Hospital Universitario La Paz, 28046 Madrid, Spain; 3Department of Nutrition, Harvard T.H. Chan School of Public Health, Boston, MA 02115, USA; 4Clinical Nutrition and Dietetics Unit, Department of Endocrinology and Nutrition, 12 de Octubre Hospital, 28041 Madrid, Spain; 5ISGlobal, Campus Mar, 08036 Barcelona, Spain; 6Unit of Cardiovascular and Nutritional Epidemiology, Institute of Environmental Medicine, Karolinska Institutet, SE-171 77 Stockholm, Sweden; 7IMDEA-Food Institute, CEI UAM+CSIC, 28049 Madrid, Spain

**Keywords:** cooking methods, anthropometrics, cardiovascular risk factors, blood pressure, cardiac function biomarkers, older adults, Seniors-ENRICA 2 cohort

## Abstract

Food consumption has a prominent role in the occurrence of cardiometabolic diseases, however, little is known about the specific influence of cooking methods. This study examined the association between cooking methods and anthropometrics, cardiovascular risk factors, and cardiac damage biomarkers in older adults. Data were taken from 2476 individuals aged ≥65 from the Seniors-ENRICA 2 cohort in Spain and recruited between 2015 and 2017. Eight cooking methods (raw, boiling, roasting, pan-frying, frying, toasting, sautéing, and stewing) were assessed using a face-to-face validated dietary history. Study associations were summarized as adjusted percentage differences (PDs) in anthropometrics, cardiovascular risk factors, and cardiac damage biomarkers between extreme sex-specific quintiles ((5th − 1st/1st) × 100) of food consumed with each cooking method, estimated using marginal effects from generalized linear models. After adjusting for potential confounders, including diet quality, PDs corresponding to raw food consumption were −13.4% (*p*-trend: <0.001) for weight, −12.9% (*p*-trend: <0.001) for body mass index (BMI), −14.8% (*p*-trend: <0.001) for triglycerides, and −13.6% (*p*-trend: <0.115) for insulin. PDs for boiled food consumption were −13.3% (*p*-trend: <0.001) for weight, −10.0% (*p*-trend: <0.001) for BMI, and −20.5% (*p*-trend: <0.001) for insulin. PDs for roasted food consumption were −11.1 (*p*-trend: <0.001) for weight and −23.3% (*p*-trend: <0.001) for insulin. PDs for pan-fried food consumption were −18.7% (*p*-trend: <0.019) for insulin, −15.3% (*p*-trend: <0.094) for pro-B-type natriuretic peptide amino-terminal, and −10.9% (*p*-trend: <0.295) for troponin T. No relevant differences were observed for blood pressure nor for other cooking methods. Raw food consumption along with boiling, roasting, and pan-frying were associated with healthier cardiovascular profiles, mainly due to lower weight and insulin levels. Future experimental research should test the effectiveness of these cooking methods for cardiovascular prevention in older adults.

## 1. Introduction

Diet is one of the leading underlying factors linked to death from chronic diseases, such as cardiovascular disease (CVD) and several types of cancer [1,2,3]. The most influential dietary guidelines, such as those for the US population [4] and other European countries [5,6], have mainly based their recommendations on nutrients, foods, and dietary patterns. However, few of them have considered the influence of cooking methods on the risk of chronic disease, in particular CVD.

Food preparation could influence original food composition. For example, by modifying nutrient and vitamin bioavailability [7], through water evaporation and its replacement with oil [8,9,10,11], alterations in antioxidant activity [12,13], formation of glycation end products [14,15,16], and modifications in glycemic and insulin responses [17,18], among others.

Evidence of the role of cooking methods on health is growing, although it is still based on a limited number of population-based studies. The main reason is the difficulty in collecting information on cooking methods when using semiquantitative food frequency questionnaires (the standard method to collect diet data in large epidemiological studies). At most, these questionnaires can identify a small group of foods that are frequently consumed fried, e.g., French fries, donuts, fried eggs, etc. [19,20,21], but data on other cooking methods is almost lacking.

Consequently, frying has been the main focus in epidemiological studies. Several studies in the US have shown a deleterious association between frying and cardiovascular health [22,23]. However, this association was not identified with mortality nor with incidence of CVD [24,25] when studies were conducted in Spain (probably due to the use of olive oil or other vegetable oils for frying in Mediterranean countries). Besides frying, other cooking methods have hardly been studied or have shown inconclusive results [26]. Nonetheless, it has been suggested that cooking methods may influence blood lipid profile [27] and CVD risk [28]. Additionally, raw (but not cooked) vegetable intake was associated with lower CVD incidence and mortality in participants from the U.K. Biobank [29], although, in these results, residual confounding could not be ruled out.

Thus, this study evaluated for the first time the association between cooking methods and anthropometrics, cardiovascular risk factors, and cardiac damage biomarkers in a large population-based sample of older adults from Spain.

## 2. Materials and Methods

### 2.1. Study Design and Participants

The Seniors-ENRICA 2 is a population-based study conducted with 3273 participants aged ≥65 and recruited from 2015 to 2017 (wave 0—baseline) in the city of Madrid and other four large adjacent cities (Getafe, Torrejón, Alcorcón, and Alcalá de Henares) [30]. Participants were selected by sex- and district-stratified random sampling based on their national healthcare card.

Information was obtained through a three-stage methodology similar to that in the Seniors-ENRICA 1 cohort [31]. Sociodemographic, lifestyle, and morbidity information was obtained by a telephone interview. Then, a nurse performed a physical examination, and blood and urine samples were collected (first home visit). Seven days later (second home visit), an interviewer obtained food consumption data and placed an ambulatory blood pressure monitor around the arm of the participant, which was removed 24 h later.

A total of 806 (24.6%) participants were excluded from the initial sample of 3273 individuals: 483 without dietary information, 12 with implausible values for total energy intake (<800 or >5000 kcal/day in men; <500 or >4000 kcal/day in women) and 9 lacking data on potential confounders, 52 on outcome variables, and 250 on other variables. Therefore, 2467 participants remained for analyses. Two laboratory parameters (glycated hemoglobin (HbA1c) and insulin) were only measured in a nested subsample consisting of 1066 participants who were recruited from January 2017 to the end of the study (Appendix A).

The Clinical Research Ethics Committee of La Paz University Hospital in Madrid approved the study protocol, and all participants provided written informed consent.

### 2.2. Diet Assessment

Habitual food consumption was collected with a validated dietary history (DH-ENRICA) conducted by trained and certified nonmedical interviewers [31]. The DH-ENRICA collects information on 861 foods consumed at least once every two weeks in the preceding year, along with its cooking technique or preservation method. A total of 127 sets of digitalized photos together with household measurements of typical Spanish foods and recipes were used for estimation of portion size [32]. The cooking methods most frequently used in Spain, with a mean consumption of at least 15 g/day, were raw, boiling, roasting, pan-frying, frying, toasting, sautéing, and stewing. Mixed cooking methods were not considered for the analyses (e.g., boiling + sautéing, frying + boiling, and sautéing + baking) due to the impossibility of distinguishing between methods. A detailed description of the main cooking methods is shown in Appendix A. Energy intake (kcal/day) as well as nutrients such as very-long-chain omega-3 fatty acids (g/day) and fiber (g/day) were derived by using standard composition tables from Spain.

### 2.3. Outcome Variables

#### 2.3.1. Anthropometrics

Trained staff measured weight, mid-upper arm circumference (MUAC), waist, hip, and calf circumferences using standardized procedures. These parameters were measured twice, and a mean between them was calculated. Body mass index (BMI) was calculated as weight divided by the square of the body height in meters (kg/m^2^). Electronic scales (model Seca 841: Seca Deutschland, Hamburg, Germany, precision to 0.1 kg), portable extendable stadiometers (model Ka We 44 444 Seca), and flexible inelastic belt-type tapes were used for these measurements.

#### 2.3.2. Cardiovascular Risk Factors

Blood samples were centrally analyzed in the CORE laboratory of La Paz University Hospital in Madrid to measure total cholesterol (colorimetric enzymatic method with cholesterol-oxidase, esterase, and peroxidase), HDL-cholesterol (direct method to eliminate other particles and its reaction with cholesterol esterase), triglycerides (colorimetric enzymatic method with lipase and glycerol kinase), and glucose (using a colorimetric hexokinase procedure). LDL-cholesterol was estimated with the Friedewald formula [33]. HbA1c (high-performance liquid chromatography) and insulin (chemiluminescent immunoassay) were also measured.

Casual blood pressure (BP) and heart rate were also measured under standardized conditions [34]. For casual BP, we used the mean from the second and third measurements. In addition, 24 h ambulatory blood pressure measurements (ABPM) were obtained using a validated non-invasive oscillometric device (Mobil-O-Graph 24 h PWA, I.E.M., Stolberg, Germany; Mediscan, Spain) programmed to register BP at 20 min intervals. Appropriate cuff sizes were used for each participant. ABPM recordings were performed on working days, and participants were instructed to maintain their usual activities and to keep the arm extended and immobile at the time of cuff inflation. ABPM were considered valid when they successfully recorded ≥70% of systolic and diastolic BPs during both daytime and nocturnal periods [35].

#### 2.3.3. Cardiac Damage

Two cardiac damage biomarkers were determined on fasting blood samples. Serum pro-B-type natriuretic peptide amino-terminal (NT-proBNP) and troponin T were measured on Cobas^®^ 6000 analyzer (Roche Diagnostics, Basel, Switzerland) by Elecsys^®^ electrochemiluminescence immunoassay. The lower detection limits were 10 and 3 pg/mL, respectively.

#### 2.3.4. Potential Confounders

Information on age, sex, educational level, smoking status, alcohol consumption, physical activity, time spent watching TV, and the number of physician-diagnosed chronic diseases was collected; we also recorded the number of prescribed medications, which were checked against the drug packages. Leisure time and household physical activity were evaluated using the EPIC short questionnaire containing 17 items. Each item was multiplied by its energy expenditure rate in Metabolic Equivalents (METS) [36], and all the activities (expressed in METS·h/week) were summed up. To adjust for body size, total energy intake was taken into account. Diet quality was represented by very-long-chain omega-3 fatty acids and fiber intake.

### 2.4. Statistical Analysis

To take into account body size and sex, food consumption was expressed as g/kg of body weight and participants were classified into sex-specific quintiles of food consumed with each cooking method. Data on anthropometrics, cardiovascular risk factors, and cardiac biomarkers were log-transformed to improve normality. Marginal effects were obtained from generalized linear models (GLM) adjusted for the potential confounders described above. Geometric means were estimated, and percentage differences (PDs) in anthropometrics, cardiovascular risk factors and cardiac damage biomarkers according to food consumed with each cooking method were calculated to summarize the study associations. We set a cut-off point of 10% for PDs to be considered clinically relevant for CVD prevention [37]. To calculate *p* for a linear trend, quintiles of food consumption were modelled as a continuous variable.

A two-tailed *p*-value < 0.05 was set as statistically significant. Analyses were performed with Stata, version 15.0 (StataCorp, College Station, TX, USA). We used the STROBE cross-sectional checklist when writing this report [38].

## 3. Results

In the study sample (*N* = 2467), the mean age of the participants was 71.6 years (SD 4.4), 53.0% were women, and most of them had primary studies or less (63.6%). This sample is also notable for a high dietary fiber intake of 32.0 g/d (SD 8.0). The most frequently consumed cooking methods were raw with 470.0 g/d (SD 208.0), boiling with 277.0 g/d (SD 119.0), roasting with 156.0 g/d (SD 68.0), and pan-frying with 63.0 g/d (SD 47.0) (Table 1).

After adjusting for potential confounders, PDs in weight and some other cardiovascular risk factors were higher than 10% for several cooking methods. Specifically, for raw food consumption, PDs between the highest and the lowest quintiles were −13.4% (*p*-trend: <0.001) for weight, −12.9% (*p*-trend: <0.001) for BMI, −14.8% (*p*-trend: <0.001) for triglycerides, and −13.6% (*p*-trend: <0.115) for insulin (Table 2). Regarding boiled food, PDs were −13.3% (*p*-trend: <0.001) for weight, −10.0% (*p*-trend: <0.001) for BMI, and −20.5% (*p*-trend: <0.001) for insulin (Table 3). PDs for roasted food were −11.1 (*p*-trend: <0.001) for weight and −23.3% (*p*-trend: <0.001) for insulin (Table 4). Finally, PDs for pan-fried food were −18.7% (*p*-trend: <0.019) for insulin (Table 5). In addition, pan-frying was the single cooking method with negative PDs for cardiac damage biomarkers: −15.3% (*p*-trend: <0.094) for NT-proBNP and −10.9 (*p*-trend: 0.295) for troponin T (Table 5).

No relevant differences were observed for other anthropometrics, blood pressure, and heart rate (Table 2, Table 3, Table 4 and Table 5) nor for other cooking methods (Appendix A).

## 4. Discussion

In this large epidemiological study, we found that four cooking methods were beneficial. Raw food consumption as well as boiling, roasting, and pan-frying (entailing no added fats exposed to high temperatures) were associated with beneficial cardiometabolic profiles. Raw food consumption, boiling, and roasting were associated with healthy weight profiles. The consumption of raw food was also associated with lower triglycerides, and these four cooking methods were associated with lower insulin levels. Finally, pan-frying was associated with lower markers of cardiac damage. According to these findings, the selection of these four cooking methods could be a strategy for cardiovascular prevention and healthy aging.

For anthropometric measurements, consumption of raw, boiled, and roasted food appeared to reduce weight and BMI. A possible explanation is that few to no added fats are used with these cooking methods [39]. In addition, in older adults, not only weight but also fat distribution is important. During aging, subcutaneous fat redistributes towards the central areas, increasing waist circumference [40] and reducing skeletal muscle mass [41]. Several studies have shown that a dietary pattern high in raw food was associated with a lower gain in waist circumference [42,43]. Our results were in line with this but without reaching clinical relevance. Small negative PDs were observed for waist circumference and for other muscle groups such as MUAC or calf circumference, which are all predictors of lower mortality in old people [44].

Differences in lipid fractions associated with cooking methods were not substantial, except for triglycerides in raw food consumption. This is partly consistent with the results of one study with 201 participants where a strict raw food diet lowered triglycerides and total cholesterol over 3.5 years of follow-up [45]. This was likely related to the low fat content of this strict diet. However, although strict raw food diets are neither common nor desirable among the general older adult population, advice to increase raw food consumption might be appropriate for individuals suffering from hypertriglyceridemia.

With regard to glucose metabolism, insulin levels were lower with higher consumption of raw, boiled, roasted, and pan-fried food. Indeed, insulin was the variable with higher negative PDs in all cooking methods. In animal models, raw starch consumption reduces fasting [46] and postprandial [47] blood glucose levels. In addition, in human studies, fresh fruit consumption decreases incident diabetes and its vascular complications in Chinese adults [48]. Aside from raw food, lower insulin levels associated with other cooking methods could be related to a higher content of water during cooking (such as in boiling) or to higher temperatures reached during the cooking process (such as in roasting and pan-frying) [49]. Additionally, these cooking methods promote resistant starch formation [50], increase insulin sensitivity [51], and improve adipose tissue metabolism [52]. Likewise, during boiling, greater resistant starch production may occur [49], which in turn, requires a longer period for digestion [53], and generates a lower glycemic load [54,55]. Therefore, we hypothesize that the four cooking methods associated with lower insulin levels might be particularly beneficial for patients with altered glycemic metabolism.

Regarding blood pressure levels, we found no strong association with the main cooking methods. In literature, the results are inconclusive [56]. Fried food consumption has been associated with a higher prevalence of elevated blood pressure in women but not men from South Korea [57], with a higher frequency of prehypertension and hypertension among Filipino women [58], and with a higher incidence of hypertension in a cohort of university graduates from Spain [59]. Except for frying, other cooking methods have barely been studied, although boiled food consumption has been proposed as a strategy to reduce hypertension [60].

Concerning cardiac damage biomarkers and its relation to cooking methods, there is little to no evidence in the literature. Based on our general results for NT-proBNP and troponin T, we suggest that pan-frying is a safe cooking method. Moreover, pan-frying with extra virgin olive oil could improve food lipid content [61]. Until there are new data from clinical trials, pan-frying with a minimum amount of oil (to prevent food sticking) might be considered a healthy cooking method.

Regarding frying, our findings are in line with the results obtained in the EPIC-Spain study, where no association was found for cardiovascular mortality or incident coronary heart disease [25] nor for stroke [24]. Thus, the results do not support a detrimental role of frying on health among older Spanish adults. One possible explanation could rely on the use of olive oil for frying in Spain, which is rich in polyphenols and antioxidants and could counteract the potentially harmful effects of frying [56]. In contrast, in the United States (where fats other than olive oil are used for frying), fried food consumption has been associated with diabetes and cardiovascular disease in the Nurses’ Health Study, in the Health Professionals Follow-up Study [23], and in the Women’s Health Initiative [22]. A further step in frying investigation was taken looking for the interaction between frying and some genes. The Women’s Genome Health Study showed a genetic predisposition for increased adiposity with the consumption of fried food [62]. In light of these findings, cautionary advice seems reasonable, emphasizing the importance of reducing fried food intake in Western countries. However, in Mediterranean countries (as in Spain), a higher risk of cardiovascular diseases associated with fried food consumption is not supported by the current evidence [56].

The study of cooking methods and their associations with cardiovascular risk factors deserves further attention. First, since there is universal exposure to them, understanding their effects on health could improve cardiometabolic profile among older adults who already have a high cardiovascular risk. Second, culinary education should be used to promote cardiovascular disease prevention [63]. For those pursuing a healthy lifestyle, simple cooking techniques could foster a culture of cooking as opposed to the consumption of ultra-processed food [64,65]. Indeed, in older adults, ultra-processed food has been associated with an increased risk of abdominal obesity [66], dyslipidemia [67], and renal function decline [68]. Finally, the selection of healthy cooking methods could derive in some benefits in specific subpopulations (e.g., individuals with overweight, obesity, hypertriglyceridemia, or among those who are prediabetics).

This study has several limitations. First, the study was cross-sectional, so causality cannot be established. However, our analyses are a first step in the assessment of cooking methods and their association with metabolic and cardiovascular risk. Second, we assessed the cooking method used and the food consumed jointly, as they were consumed, which did not allow us to distinguish between them. Third, participants used colloquial expressions to report cooking methods, and some degree of misclassification cannot be ruled out. Nonetheless, this information allows simple advice on cooking methods to be disseminated and understood by the general population. Finally, the sample was not representative of the adult Spanish population since data collection was only conducted in the Community of Madrid, and cardiovascular risk factors might vary throughout the country. Nevertheless, the sample was population-based, selected through random sampling, and participants came from both urban and rural areas. Furthermore, there might be geographical differences in the use of cooking methods throughout Spain; however, age is the variable that explains most of these differences [69]. Among the study strengths is the use of a dietary history, which takes into account the recipes usually cooked in Spain and was shown to be valid and reproducible in the Spanish population. Additionally, we used standardized procedures to collect anthropometric data, as well as casual and 24 h blood pressure. Likewise, all biological samples were analyzed in a central laboratory to ensure reliability. Furthermore, several confounding factors were controlled for, although some residual confounding cannot be ruled out. Lastly, this is the first time in which these relationships have been examined in a large epidemiological study.

## 5. Conclusions

In conclusion, some cooking methods, such as raw, boiling, roasting, and pan-frying, were associated with a healthier cardiometabolic profile. These results highlight the importance of cooking methods and their potential benefits in cardiovascular prevention. More studies are needed to establish causality and to fully understand the impact of cooking methods on health.

## Figures and Tables

**Table 1 nutrients-14-03426-t001:** Characteristics of the participants in the Seniors ENRICA 2 study (N = 2467).

Characteristics	Total
N	2467
Women, (%)	1308 (53.0)
Age, mean (SD), years	71.6 (4.4)
Educational level, no. (%)	
Primary or less	1569 (63.6)
Secondary	460 (18.7)
Higher	438 (17.8)
Cigarette smoking status, no. (%)	
Current	226 (9.2)
Former	941 (38.1)
Never	1300 (52.7)
Alcohol consumption, median (IQR), g/d	
Ex-drinker status, no. (%)	251 (10.2)
Recreational physical activity, median (IQR), METS·h/week	24.5 (16.3–36.8)
Household physical activity, median (IQR), METS·h/week	35.0 (17.5–54.6)
Hours of television watching, mean (SD)	22.3 (11.0)
No. of chronic diseases *, median (IQR)	1.0 (0–2.0)
No. of medications, median (IQR)	3.0 (1.0–5.0)
Dietary variables	
Energy intake, mean (SD), kcal/d	2382 (449)
Very-long-chain omega-3 fatty acids, median (IQR), g/d	0.6 (0.3–0.9)
Fiber, mean (SD), g/d	31.5 (8.4)
Cooking methods consumption	
Raw, mean (SD), g/d	470 (208)
Boiling, mean (SD), g/d	277 (119)
Roasting, mean (SD), g/d	156 (68.0)
Pan-frying, mean (SD), g/d	63.0 (47.0)
Frying, mean (SD), g/d	42.0 (33.0)
Toasting, mean (SD), g/d	42.0 (40.0)
Sautéing, mean (SD), g/d	22.0 (21.0)
Stewing, mean (SD), g/d	19.0 (21.0)

SD: standard deviation; IQR: interquartile range; METS: metabolic equivalent of task. * Chronic diseases: pneumonia, asthma, cardiac infarction, stroke, heart failure, atrial fibrillation, arthrosis, rheumatoid arthritis, hip fracture, cholelithiasis, cirrhosis, urinary tract infections, cataract, depression, anxiety, Parkinson disease, dementia/Alzheimer, periodontal disease, obstructive sleep apnea, and cancer.

**Table 2 nutrients-14-03426-t002:** Adjusted means (95% confidence interval), percentage difference, and *p* for a linear trend across quintiles of raw food consumption.

Raw Food Consumption							
	Q1	Q2	Q3	Q4	Q5	PDs	*p*
n (median, g/kg of body weight) in men	232 (2.98)	232 (4.90)	232 (6.32)	232 (8.08)	231 (10.5)		
n (median, g/kg of body weight) in women	262 (3.15)	262 (4.77)	261 (6.30)	262 (8.02)	261 (10.9)		
Anthropometrics							
Weight (kg)	76.7 (76.1–77.3)	74.2 (73.6–74.9)	71.6 (71.0–72.2)	70.4 (69.8–71.0)	66.4 (65.8–67.0)	−13.4	<0.001
BMI (kg/m^2^)	29.4 (29.2–29.6)	28.7 (28.5–28.8)	27.7 (27.5–27.8)	27.3 (27.1–27.4)	25.6 (25.5–25.8)	−12.9	<0.001
MUAC (cm)	29.3 (29.2–29.4)	28.9 (28.8–29.0)	28.8 (28.7–28.9)	28.4 (28.3–28.5)	27.7 (27.6–27.8)	−5.50	<0.001
Waist circumference (cm)	100.3 (99.8–100.8)	97.9 (97.3–98.4)	95.3 (94.8–95.8)	94.8 (94.2–95.3)	91.5 (91.0–92.0)	−8.80	<0.001
Hip circumference (cm)	106.2 (105.9–106.5)	104.2 (103.9–104.5)	102.6 (102.3–102.9)	101.9 (101.6–102.2)	99.7 (99.4–100)	−6.10	<0.001
Calf circumference (cm)	34.4 (34.3–34.5)	34.1 (34.0–34.2)	33.6 (33.6–33.7)	33.6 (33.5–33.7)	33.2 (33.1–33.3)	−3.50	<0.001
Cardiovascular risk factors							
Total cholesterol (mg/dL)	190.3 (188.9–191.8)	189.2 (187.7–190.7)	188.3 (186.8–189.8)	189.7 (188.2–191.1)	192.4 (190.9–193.9)	1.10	0.949
HDL-cholesterol (mg/dL)	53.1 (52.5–53.6)	52.1 (51.6–52.7)	53.5 (52.9–54.1)	53.9 (53.3–54.5)	57.8 (57.2–58.4)	8.90	<0.001
LDL-cholesterol (mg/dL)	113.6 (112.6–114.7)	113.5 (112.4–114.7)	112.6 (111.5–113.7)	114.0 (112.9–115.0)	114.4 (113.4–115.5)	0.70	0.409
Triglycerides (mg/dL)	110.6 (109.7–111.5)	111.5 (110.5–112.4)	104.6 (103.7–105.5)	102.4 (101.6–103.2)	94.2 (93.5–94.9)	−14.8	<0.001
Glucose (mg/dL)	100.3 (99.7–100.9)	100.3 (99.7–100.9)	97.0 (96.5–97.6)	97.0 (96.5–97.6)	93.7 (93.2–94.2)	−6.60	<0.001
HbA1c (%)	5.82 (5.79–5.85)	5.89 (5.86–5.92)	5.75 (5.72–5.78)	5.77 (5.74–5.80)	5.70 (5.67–5.72)	−2.10	0.086
Insulin (μU/mL)	10.6 (10.3–10.9)	10.9 (10.7–11.2)	9.05 (8.81–9.29)	9.89 (9.64–10.1)	9.16 (8.96–9.37)	−13.6	0.115
Blood pressure							
Casual SBP (mmHg)	136.1 (135.8–136.3)	137.0 (136.8–137.2)	134.5 (134.3–134.7)	135.5 (135.3–135.7)	134.9 (134.7–135.1)	−0.90	0.092
Casual DBP (mmHg)	79.8 (79.5–80.1)	81.1 (80.8–81.4)	79.2 (78.9–79.5)	80.1 (79.9–80.4)	79.8 (79.6–80.1)	0.00	0.325
Casual HR (bpm)	71.2 (71.0–71.5)	71.0 (70.8–71.2)	70.3 (70.1–70.5)	69.1 (68.8–69.3)	69.2 (69.0–69.5)	−2.80	0.058
24 h SBP (mmHg)	128.4 (128.2–128.6)	128.1 (127.9–128.3)	126.6 (126.4–126.8)	127.2 (127.0–127.4)	125.3 (125.1–125.4)	−2.40	0.005
24 h DBP (mmHg)	74.9 (74.6–75.1)	75.0 (74.7–75.2)	74.0 (73.7–74.2)	74.5 (74.3–74.7)	74.0 (73.8–74.2)	−1.20	0.148
24 h HR (bpm)	69.4 (69.2–69.6)	68.3 (68.1–68.5)	68.0 (67.9–68.5)	67.0 (66.8–67.2)	67.1 (66.9–67.3)	−3.30	0.028
Cardiac function biomarkers							
NT-proBNP (pg/mL)	92.9 (90.3–95.6)	81.9 (79.7–84.2)	82.2 (80.1–84.3)	87.0 (84.7–89.3)	88.1 (86.0–90.3	−5.20	0.649
Troponin T (ng/L)	10.0 (9.80–10.2)	9.90 (9.70–10.1)	9.30 (9.10–9.50)	9.40 (9.20–9.60)	9.50 (9.30–9.70)	−5.00	0.287

Analyses were adjusted for sex, age (continuous), educational level (primary or less, secondary, university), cigarette smoking status (former, current, never), alcohol consumption (continuous), ex-drinker status (yes, no), recreational physical activity in METS·hours/week (continuous), household physical activity in METS·hours/week (continuous), hours of television (continuous), number of chronic diseases (continuous), number of prescribed medications (continuous), energy intake in kcal/d (continuous), very-long-chain omega-3 fatty acids consumption (continuous), and fiber consumption (continuous). PDs: percentage difference calculated as [(5th quintile − 1st quintile)/1st quintile] × 100, *p*: *p* for a linear trend. BMI: body mass index, MUAC: mid-upper arm circumference, SBP: systolic blood pressure, DBP: diastolic blood pressure, HR: heart rate. Analyses for HbA1c and insulin were performed in a nested subsample of 1066 participants.

**Table 3 nutrients-14-03426-t003:** Adjusted means (95% confidence interval), percentage difference, and *p* for a linear trend across quintiles of boiled food consumption.

Boiled Food Consumption							
	Q1	Q2	Q3	Q4	Q5	PDs	*p*
n (median, g/kg of body weight) in men	232 (2.04)	232 (2.91)	232 (3.62)	232 (4.59)	231 (6.09)		
n (median, g/kg of body weight) in women	262 (2.02)	262 (2.90)	261 (3.65)	262 (4.57)	261 (6.10)		
Anthropometrics							
Weight (kg)	76.9 (76.3–77.5)	74.0 (73.3–74.6)	72.0 (71.4–72.6)	69.9 (69.2–70.6)	66.7 (65.9–67.4)	−13.3	<0.001
BMI (kg/m^2^)	29.1 (29.0–29.3)	28.2 (28.1–28.4)	27.9 (27.7–28.1)	27.2 (27.0–27.3)	26.2 (26.0–26.4)	−10.0	<0.001
MUAC (cm)	29.2 (29.1–29.3)	28.9 (28.8–28.9)	28.6 (28.6–28.7)	28.4 (28.3–28.5)	27.9 (27.8–28.0)	−4.50	<0.001
Waist circumference (cm)	99.3 (98.8–99.8)	97.7 (97.1–98.2)	96.4 (95.9–97.0)	94.4 (93.8–95.0)	91.8 (91.2–92.4	−7.60	<0.001
Hip circumference (cm)	105.6 (105.3–106.0)	103.9 (103.6–104.2)	103.1 (102.8–103.4)	101.5 (101.2–101.8)	100.4 (100.1–100.7)	−4.90	<0.001
Calf circumference (cm)	34.3 (34.3–34.4)	34.0 (33.9–34.1)	33.7 (33.6–33.8)	33.7 (33.6–33.8)	33.2 (33.1–33.3)	−3.20	<0.001
Cardiovascular risk factors							
Total cholesterol (mg/dL)	188.6 (187.2–190.1)	189.2 (187.7–190.6)	190.0 (188.5–191.5)	190.9 (189.4–192.3)	191.3 (189.8–192.8)	1.40	0.021
HDL-cholesterol (mg/dL)	52.6 (52.0–53.2)	53.8 (53.2–54.3)	54.4 (53.8–55.0)	54.6 (54.0–55.2)	55.0 (54.4–55.7)	4.60	<0.001
LDL-cholesterol (mg/dL)	112.6 (111.6–113.7)	113.7 (112.6–114.8)	113.5 (112.4–114.6)	113.8 (112.8–114.9)	114.5 (113.4–115.5)	1.70	0.152
Triglycerides (mg/dL)	109.2 (108.2–110.3)	102.0 (101.1–102.9)	103.8 (102.9–104.7)	105.3 (10.4–106.3)	102.1 (101.3–103.0)	−6.50	0.043
Glucose (mg/dL)	98.4 (97.8–99.0)	97.2 (96.6–97.8)	98.5 (97.9–99.1)	97.7 (97.1–98.3)	96.4 (95.8–97.0)	−2.00	0.214
HbA1c (%)	5.82 (5.79–5.85)	5.76 (5.73–5.78)	5.83 (5.80–5.86)	5.74 (5.71–5.77)	5.77 (5.74–5.80)	−0.90	0.338
Insulin (μU/mL)	11.0 (10.7–11.3)	9.94 (9.70–10.2)	10.1 (9.80–10.4)	9.63 (9.34–9.92)	8.75 (8.52–8.99)	−20.5	<0.001
Blood pressure							
Casual SBP (mmHg)	135.0 (134.8–135.1)	135.9 (135.7–136.1)	135.4 (135.2–135.6)	135.4 (135.2–135.6)	136.3 (136.1–136.5)	1.00	0.884
Casual DBP (mmHg)	79.4 (79.1–79.7)	80.2 (79.9–80.4)	80.0 (79.7–80.2)	80.0 (79.7–80.3)	80.5 (80.2–80.8)	1.40	0.539
Casual HR (bpm)	69.8 (69.5–70.0)	69.6 (69.3–69.8)	70.5 (70.3–70.8)	70.7 (70.5–70.9)	70.3 (70.0–70.5)	0.70	0.042
24 h SBP (mmHg)	127.5 (127.3–127.7)	128.0 (127.8–128.2)	127.2 (127.1–127.4)	125.9 (125.7–126.1)	127.0 (126.8–127.1)	−0.40	0.084
24 h DBP (mmHg)	74.7 (74.4–74.9)	75.0 (74.7–75.2)	74.3 (74.1–74.5)	74.0 (73.7–74.2)	74.4 (74.1–74.6)	−0.40	0.085
24 h HR (bpm)	68.0 (67.8–68.3)	67.7 (67.5–67.9)	68.6 (68.4–68.9)	67.9 (67.6–68.1)	67.6 (67.4–67.9)	−0.60	0.495
Cardiac function biomarkers							
NT-proBNP (pg/mL)	83.5 (81.5–85.6)	86.2 (84.0–88.5)	90.1 (87.7–92.5)	83.0 (80.9–85.2)	89.1 (86.7–91.7)	6.70	0.142
Troponin T (ng/L)	9.90 (9.70–10.0)	9.70 (9.50–9.90)	9.60 (9.40–9.90)	9.30 (9.10–9.50)	9.50 (9.30–9.70)	−4.00	0.032

Analyses were adjusted for sex, age (continuous), educational level (primary or less, secondary, university), cigarette smoking status (former, current, never), alcohol consumption (continuous), ex-drinker status (yes, no), recreational physical activity in METS·hours/week (continuous), household physical activity in METS·hours/week (continuous), hours of television (continuous), number of chronic diseases (continuous), number of prescribed medications (continuous), energy intake in kcal/d (continuous), very-long-chain omega-3 fatty acids consumption (continuous), and fiber consumption (continuous). PDs: percentage difference calculated as [(5th quintile − 1st quintile)/1st quintile] × 100, *p*: *p* for a linear trend. BMI: body mass index, MUAC: mid-upper arm circumference, SBP: systolic blood pressure, DBP: diastolic blood pressure, HR: heart rate. Analyses for HbA1c and insulin were performed in a nested subsample of 1066 participants.

**Table 4 nutrients-14-03426-t004:** Adjusted means (95% confidence interval), percentage difference, and *p* for a linear trend across quintiles of roasted food consumption.

	Roasted Food Consumption						
	Q1	Q2	Q3	Q4	Q5	PC	*p*
n (median, g/kg of body weight) in men	232 (1.06)	232 (1.72)	232 (2.15)	232 (2.58)	232 (3.41)		
n (median, g/kg of body weight) in women	262 (0.960)	262 (1.70)	261 (2.16)	262 (2.65)	261 (3.49)		
Anthropometrics							
Weight (kg)	73.8 (73.1–74.4)	75.8 (75.2–76.5)	73.2 (72.6–73.9)	71.0 (70.4–71.6)	65.6 (64.9–66.2)	−11.1	<0.001
BMI (kg/m^2^)	28.3 (28.1–28.5)	28.9 (28.7–29.0)	28.2 (28.1–28.4)	27.5 (27.3–27.6)	25.8 (25.6–25.9)	−8.80	<0.001
MUAC (cm)	29.2 (29.1–29.3)	29.5 (29.4–29.6)	28.6 (28.5–28.7)	28.5 (28.4–28.6)	27.2 (27.1–27.3)	−6.90	<0.001
Waist circumference (cm)	96.7 (96.1–97.2)	99.3 (98.7–99.8)	97.1 (96.5–97.6)	95.5 (95.0–96.0)	91.1 (90.6–91.7)	−5.80	<0.001
Hip circumference (cm)	104.1 (103.8–104.4)	105.3 (105.0–105.6)	103.5 (103.2–103.8)	102.7 (102.4–103.0)	98.9 (98.6–99.2)	−5.20	<0.001
Calf circumference (cm)	34.4 (34.3–34.5)	34.4 (34.4–34.5)	34.0 (34.0–34.1)	33.8 (33.7–33.9)	32.3 (32.2–32.4)	−6.10	<0.001
Cardiovascular risk factors							
Total cholesterol (mg/dL)	190.8 (189.2–192.3)	188.5 (187.0–190.0)	188.9 (187.5–190.4)	191.2 (189.8–192.6)	190.6 (189.1–192.0)	−0.10	0.622
HDL-cholesterol (mg/dL)	54.2 (53.6–54.8)	52.1 (51.5–52.7)	53.6 (53.0–54.2)	54.8 (54.2–55.4)	55.6 (55.0–56.2)	2.60	<0.001
LDL-cholesterol (mg/dL)	114.3 (113.1–115.4)	113.1 (112.0–114.2)	112.6 (111.5–113.6)	114.5 (113.5–115.5)	113.7 (112.6–114.8)	−0.50	0.817
Triglycerides (mg/dL)	103.6 (102.7–104.5)	108.9 (107.9–110.0)	106.2 (105.3–107.1)	103.0 (102.1–103.9)	100.8 (99.9–101.7)	−2.70	0.030
Glucose (mg/dL)	97.1 (96.4–97.7)	98.7 (98.1–99.3)	98.6 (98.0–99.2)	97.1 (96.6–97.7)	96.7 (96.1–97.2)	−0.40	0.300
HbA1c (%)	5.77 (5.74–5.80)	5.84 (5.81–5.87)	5.82 (5.79–5.85)	5.73 (5.70–5.75)	5.74 (5.71–5.77)	−0.50	0.163
Insulin (μU/mL)	10.7 (10.4–11.0)	10.4 (10.1–10.7)	10.5 (10.2–10.8)	9.21 (9.00–9.44)	8.21 (7.96–8.46)	−23.3	<0.001
Blood pressure							
Casual SBP (mmHg)	135.0 (134.8–135.2)	135.4 (135.2–135.6)	136.2 (136.0–136.4)	135.0 (134.8–135.2)	136.4 (136.2–136.6)	−1.04	0.819
Casual DBP (mmHg)	79.7 (79.4–78.0)	80.4 (80.1–80.6)	80.1 (79.8–80.3)	79.7 (79.4–80.0)	80.2 (79.9–80.5)	0.60	0.803
Casual HR (bpm)	70.1 (69.9–70.4)	70.3(70.1–70.5)	69.7 (69.5–69.9)	70.7 (70.5–70.9)	70.0 (69.8–70.2)	−0.10	0.751
24 h SBP (mmHg)	126.8 (126.6–126.9)	127.8 (127.6–127.9)	127.4 (127.3–127.6)	126.7 (126.6–126.9)	126.9 (126.7–127.0)	0.10	0.210
24 h DBP (mmHg)	74.4 (74.2–74.6)	74.9 (74.7–75.1)	74.7 (74.5–74.9)	74.0 (73.8–74.3)	74.2 (74.0–74.4)	−0.30	0.138
24 h HR (bpm)	67.9 (67.7–68.1)	68.0 (67.7–68.2)	67.9 (67.6–68.1)	68.4 (68.2–68.7)	67.8 (67.5–68.0)	−0.10	0.922
Cardiac function biomarkers							
NT-proBNP (pg/mL)	84.3 (82.1–86.5)	81.7 (79.6–83.9)	89.9 (87.7–92.2)	84.4 (82.1–86.7)	91.8 (89.4–94.3)	8.90	0.191
Troponin T (ng/L)	9.20 (9.00–9.40)	9.80 (9.60–10.0)	9.80 (9.60–10.0)	9.50 (9.30–9.70)	9.80 (9.60–10.0)	6.50	0.751

Analyses were adjusted for sex, age (continuous), educational level (primary or less, secondary, university), cigarette smoking status (former, current, never), alcohol consumption (continuous), ex-drinker status (yes, no), recreational physical activity in METS·hours/week (continuous), household physical activity in METS·hours/week (continuous), hours of television (continuous), number of chronic diseases (continuous), number of prescribed medications (continuous), energy intake in kcal/d (continuous), very-long-chain omega-3 fatty acids consumption (continuous), and fiber consumption (continuous). PDs: percentage difference calculated as [(5th quintile − 1st quintile)/1st quintile] × 100, *p*: *p* for a linear trend. BMI: body mass index, MUAC: mid-upper arm circumference, SBP: systolic blood pressure, DBP: diastolic blood pressure, HR: heart rate. Analyses for HbA1c and insulin were performed in a nested subsample of 1066 participants.

**Table 5 nutrients-14-03426-t005:** Adjusted means (95% confidence interval), percentage difference, and *p* for a linear trend across quintiles of pan-fried food consumption.

	Pan-Fried Food Consumption						
	Q1	Q2	Q3	Q4	Q5	PC	*p*
n (median, g/kg of body weight) in men	232 (0.200)	232 (0.480)	232 (0.760)	232 (1.13)	231 (1.74)		
n (median, g/kg of body weight) in women	262 (0.130)	262 (0.500)	261 (0.770)	262 (1.14)	261 (1.78)		
Anthropometrics							
Weight (kg)	73.7 (73.1–74.4)	72.9 (72.3–73.5)	72.4 (71.8–73.0)	71.4 (70.7–72.0)	69.0 (68.4–69.6)	−6.40	<0.001
BMI (kg/m^2^)	28.5 (28.3–28.6)	28.1 (27.9–28.2)	27.7 (27.6–27.9)	27.6 (27.4–27.8)	26.8 (26.6–26.9)	−6.00	<0.001
MUAC (cm)	28.8 (28.7–28.9)	28.9 (28.8–29.0)	28.5 (28.4–28.6)	28.6 (28.5–28.7)	28.2 (28.1–28.3)	−2.10	0.026
Waist circumference (cm)	97.7 (97.2–98.3)	96.3 (95.8–96.8)	96.3 (95.8–96.8)	95.3 (94.8–95.9)	94.0 (93.5–94.5)	−3.80	<0.001
Hip circumference (cm)	104.3 (104.0–104.7)	103.6 (103.3–103.9)	102.9 (102.6–103.2)	102.6 (102.3–102.9)	101.2 (100.9–101.5)	−3.00	<0.001
Calf circumference (cm)	33.8 (33.7–33.9)	34.2 (34.1–34.3)	33.8 (33.7–3.9)	33.8 (33.8–33.9)	33.4 (33.3–33.5)	−1.20	0.013
Cardiovascular risk factors							
Total cholesterol (mg/dL)	190.3 (188.8–191.7)	191.1 (189.6–192.5)	90.6 (189.2–192.0)	188.4 (186.9–189.9)	189.6 (188.1–191.1)	−0.40	0.157
HDL-cholesterol (mg/dL)	54.2 (53.6–54.8)	53.8 (53.2–54.4)	53.8 (53.2–54.4)	53.6 (53.0–54.1)	55.0 (54.4–55.6)	1.50	0.939
LDL-cholesterol (mg/dL)	113.3 (112.2–114.4)	114.6 (113.8–115.9)	114.8 (113.8–115.9)	112.2 (111.1–113.3)	113.3 (112.2–114.4)	0.00	0.216
Triglycerides (mg/dL)	107.1 (106.1–108.1)	107.0 (106.1–107.9)	102.9 (102.1–103.8)	105.2 (104.3–106.1)	100.2 (99.4–101.0)	−6.40	0.141
Glucose (mg/dL)	97.9 (97.3–98.5)	98.4 (97.8–99.0)	97.2 (96.6–97.8)	97.7 (97.1–98.3)	96.9 (96.4–97.5)	−1.02	0.955
HbA1c (%)	5.79 (5.75–5.82)	5.78 (5.75–5.80)	5.78 (5.75–5.81)	5.77 (5.74–5.80)	5.79 (5.76–5.81)	0.00	0.194
Insulin (μU/mL)	10.6 (10.3–10.9)	10.3 (10.1–10.6)	9.87 (9.62–10.1)	10.1(9.88–10.4)	8.62 (8.44–8.82)	−18.7	0.019
Blood pressure							
Casual SBP (mmHg)	135.3 (135.1–135.5)	136.1 (136.0–136.3)	136.0 (135.8–136.2)	136.2 (136.0–136.4)	134.3 (134.1–134.5)	−0.70	0.655
Casual DBP (mmHg)	80.0 (79.7–80.2)	80.5 (80.3–80.8)	80.4 (80.1–80.6)	80.4 (80.1–80.7)	78.8 (78.5–79.0)	−1.50	0.112
Casual HR (bpm)	71.4 (71.2–71.6)	69.6 (69.3–69.8)	70.2 (70.0–70.4)	69.9 (69.6–70.1)	69.8 (69.6–70.0)	−2.20	0.610
24 h SBP (mmHg)	127.1 (126.9–127.3)	127.6 (127.5–127.8)	127.2 (127.1–127.4)	128.0 (127.8–128.2)	125.6 (125.4–125.7)	−1.20	0.862
24 h DBP (mmHg)	74.4 (74.1–74.6)	74.7 (74.4–74.9)	74.8 (74.6–75.0)	75.1 (74.9–75.4)	73.3 (73.1–73.6)	−1.50	0.251
24 h HR (bpm)	68.5 (68.5–69.0)	67.2 (67.0–67.4)	68.2 (67.9–68.4)	68.2 (68.0–68.4)	67.6 (67.4–67.8)	−1.30	0.875
Cardiac function biomarkers							
NT-proBNP (pg/mL)	94.9 (92.1–97.8)	83.7 (81.7–85.8)	88.3 (86.1–90.5)	85.0 (82.9–87.2)	80.4 (78.4–82.6)	−15.3	0.094
Troponin T (ng/L)	10.1 (9.90–10.3)	9.40 (9.20–9.60)	9.70 (9.50–9.90)	9.90 (9.70–10.1)	9.00 (8.80–9.20)	−10.9	0.295

Analyses were adjusted for sex, age (continuous), educational level (primary or less, secondary, university), cigarette smoking status (former, current, never), alcohol consumption (continuous), ex-drinker status (yes, no), recreational physical activity in METS·hours/week (continuous), household physical activity in METS·hours/week (continuous), hours of television (continuous), number of chronic diseases (continuous), number of prescribed medications (continuous), energy intake in kcal/d (continuous), very-long-chain omega-3 fatty acids consumption (continuous), and fiber consumption (continuous). PDs: percentage difference calculated as [(5th quintile − 1st quintile)/1st quintile] × 100, *p*: *p* for a linear trend. BMI: body mass index, MUAC: mid-upper arm circumference, SBP: systolic blood pressure, DBP: diastolic blood pressure, HR: heart rate. Analyses for HbA1c and insulin were performed in a nested subsample of 1066 participants.

## Data Availability

The datasets generated and/or analyzed in the current study are available from the corresponding author on reasonable request.

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
