# Peer review of "Cooking Methods and Their Relationship with Anthropometrics and Cardiovascular Risk Factors among Older Spanish Adults"

_nutrients, 2022, doi:10.3390/nu14163426_

Round 1
Reviewer 1 Report
Summary: The authors were able to utilize a large sample size from a cohort of individuals >/= 65 years of age and analyze survey, interview, and physical examination data to assess the impact of eight different cooking methods on cardiovascular risk factors. They determined that certain methods (boiling, roasting, and pan-frying) along with raw food consumption were associated with healthier cardiovascular profiles.
Comments:
1). As this study was conducted in region of Madrid, are these findings really only application to individuals in this region, or are cooking styles as well as CVD risk in Spain highly variable depending on region? Can a comment addressing this be made in the Discussion?
2). Though education level of the subjects was reported, is there any information about mean household income? Or due to age, are many of this individuals on fixed incomes (i.e. pensioners)?
3). Overall, this population appears to be generally very healthy as demonstrated by the lack of chronic medical conditions? Do the authors contend that cooking methods assessed have been part of the subjects' lifestyle over a longer period of time (ie. decades), or relatively recent? If some of the methods are going to be promoted as more beneficial than other based on these results, can it be assumed that starting at a younger age is ideal?
4) Is olive the main fat used for pan-frying among these subjects?
Reviewer 2 Report
Rodríguez-Ayala et al. evaluated the relationship between 8 cooking methods and anthropometrics and cardiovascular risk factors in Spanish older adults from Seniors-ENRICA 2 cohort. The research is interesting. I have some suggestions for this study.
1. How to evaluate and calculate the quintiles of food consumption should be described clearly in the method section.
2. In this study, only descriptive results were demonstrated. The researchers mentioned p-trend results after adjusting for potential confounders using generalized linear models. How did the researcher select and evaluate the potential confounders in this study?
3. The author used “adjusted” means terminology in each table. What does it mean?
4. The missing value problems were not mentioned in this study. Did everyone have 24 hours blood pressure recorded in this study?
5. A error typing was identified. In Table 2, the hip circumference Q2 should be “104.2”. The casual SBP Q2 should be “137.0”. The casual 24h-SBP should be “128.1”.
6. Finally, it will be interesting to present the association between cooking methods and metabolic syndrome (by definition), insulin resistance (if available), and a 10-year cardiovascular risk score (Framingham risk score, SCORE2, or ASCVD Risk).
Round 2
Reviewer 2 Report
All comments had been revised accordingly. I have no further suggestions.